# Recent Progress on Hydrogel-Based Piezoelectric Devices for Biomedical Applications

**DOI:** 10.3390/mi14010167

**Published:** 2023-01-09

**Authors:** Yuxuan Du, Wenya Du, Dabin Lin, Minghao Ai, Songhang Li, Lin Zhang

**Affiliations:** 1Department of Materials Science, University of Southern California, Los Angeles, CA 90018, USA; 2Media Lab, Massachusetts Institute of Technology, Cambridge, MA 02139, USA; 3Shaanxi Province Key Laboratory of Thin Films Technology and Optical Test, School of Optoelectronic Engineering, Xi’an Technological University, Xi’an 710032, China; 4College of Engineering and Computer Science, Syracuse University, Syracuse, NY 13202, USA; 5Department of Physics and Astronomy, Franklin & Marshall College, Lancaster, PA 17604, USA

**Keywords:** piezoelectric materials, hydrogel, composites, sensors, energy harvesting, stimulation, wound healing, ultrasound

## Abstract

Flexible electronics have great potential in the application of wearable and implantable devices. Through suitable chemical alteration, hydrogels, which are three-dimensional polymeric networks, demonstrate amazing stretchability and flexibility. Hydrogel-based electronics have been widely used in wearable sensing devices because of their biomimetic structure, biocompatibility, and stimuli-responsive electrical properties. Recently, hydrogel-based piezoelectric devices have attracted intensive attention because of the combination of their unique piezoelectric performance and conductive hydrogel configuration. This mini review is to give a summary of this exciting topic with a new insight into the design and strategy of hydrogel-based piezoelectric devices. We first briefly review the representative synthesis methods and strategies of hydrogels. Subsequently, this review provides several promising biomedical applications, such as bio-signal sensing, energy harvesting, wound healing, and ultrasonic stimulation. In the end, we also provide a personal perspective on the future strategies and address the remaining challenges on hydrogel-based piezoelectric electronics.

## 1. Introduction

The growing requirement for the incorporation of electronic technology into flexible and wearable devices is being driven by the necessity of conducting individual monitoring, which is both continuous and long-term, at any given location and at any given time [1,2]. This demand is being pushed forward by the fact that the concept of continuous monitoring is becoming more and more significant. For next-generation applications in a variety of fields, such as health monitoring, smart communication, green electronics, and artificial intelligence systems, many electronic devices have been developed with flexible configurations and high performance [3,4,5].

Intelligent and adaptable electrical components are needed for wearable health-monitoring systems. Due to their responsiveness to external stimuli like stress, light, temperature, moisture or pH, and electric or magnetic fields, intelligent materials, also known as smart or responsive materials, play an important role in our lives. They cover a wide range of sensing and actuation applications in health care. The motions of people, animals, and the environment can all be exploited to generate biomechanical energy in these materials. Piezoelectric materials have drawn a lot of attention in comparison to other smart materials because of their high sensitivity, high resonance frequency, and high stability, which are crucial for electronic devices such as actuators, sensors, accelerators, ultrasonic motors, transducers, filters, and resonators, as well as micro-electromechanical systems (MEMS). Electronics built on the piezoelectric principle have emerged as the most appealing and popular functional materials and systems [6,7]. Piezoelectric materials are a family of materials that can transform mechanical force into electricity and vice versa [8,9]. The presence of piezoelectricity in inorganic piezoelectric crystals can be attributed to the configuration of ions within the non-inversely symmetric structure of the dielectric material [10,11]. The internal polarization of the material shifts in a linear fashion in response to the applied stress, which results in the development of an electrical field at the boundary of the material [6,12,13,14]. In organic piezoelectric polymers, the piezoelectric effect is created by the orientation of the polymer’s molecules, which in turn is influenced by the molecular structure of the polymer [15,16].

Piezoelectric sensors may generate their own power by the application of mechanical stimulation, in contrast to more conventional sensors, which need an external power supply to function [17]. Piezoelectric energy harvesters are a promising contender for self-powered bio-applications since they have the ability to transform mechanical energy into useable electrical energy at the nanoscale scale [18]. As a result, piezoelectric energy harvesters also provide flexibility, superior biocompatibility, compact size, high electrical output, and universal adaptability. These piezoelectric properties are crucial for a variety of bio-applications, including implantable medical devices, bioMEMS, and medical implants. Without the need for particular manufacturing, piezoelectric devices are a great option for achieving any in vivo or ex vivo biomedical application. For human motion detection, biomedical robotics, animal sensory systems, and health-monitoring systems, flexible, self-powered, miniature, ultrasensitive flow sensors are in high demand [19]. Furthermore, in recent years, bio-inspired self-powered piezoelectric electronics have received a lot of attention [20]. Different organisms in nature have evolved over time to develop highly accurate, miniaturized, and sensitive sensing systems [21]. Scientists are also continuously taking inspiration from nature to create novel artificial sensors using existing technologies. Although the performance does not match that of natural sensors from natural organisms, there have been continuous attempts to create high-performance sensors [21,22].

Both a natural strong barrier and a very sensitive sensory receptor, human skin is able to both protect the body from damage from the outside and sense a variety of minute external inputs [23]. Human skin is a biological tissue organ that is soft, stretchy, water-rich, and breathable. To meet the requirements of flexibility and conformability from human skin, many soft piezoelectric devices with different stretchable substrates have been investigated, including silicon elastomers [24,25], conducting polymers [26,27], textiles [28], and aerogels [29]. However, the majority of synthetic skin materials (such as elastomers and textiles) reported thus far have high moduli, are water- and airtight, and do not resemble the physical and mechanical characteristics of human skin.

Research in the developing topic of flexible electronics draws from a wide variety of fields, including but not limited to physics, chemistry, materials science, electrical engineering, and biology. However, the widespread application of flexible electronics is still limited because of a number of restrictions, the most significant of which are a high Young’s modulus, poor biocompatibility, and low response [30]. Numerous artificial skin-like electronic materials have been investigated for use in soft robots, artificial intelligence, biomedical prosthesis, and wearable electronics devices [31]. These materials were inspired by these characteristics. However, there are still significant difficulties in replicating the special biological makeup and sensory capabilities of human skin. It would be extremely beneficial if innovative materials could be developed with the goal of eliminating these disadvantages and expanding their practical applications. Because of its outstanding material properties, hydrogel is a family of three-dimensional cross-linked hydrated polymer networks. Because of these qualities, hydrogel is a promising contender for the next generation of flexible electronics [32].

In recent years, there has been an increase in the utilization of piezoelectric electronics and hydrogel in wearable technology. Piezoelectric materials and components should possess highly desirable qualities such as softness, deformability, and biocompatibility to achieve smooth interface with biological tissues. This has resulted in a growing interest in hydrogels because they are rich in water, have a soft mechanical structure, and are biocompatible; these qualities are similar to those of real biological tissues [33]. When used as device substrates, hydrogels’ biocompatibility, stretchability, and ductility make it possible for devices to be attached to the human body in a more straightforward manner. Traditional substrates of piezoelectric devices, on the other hand, lack the ductility and stretchability that are essential for wearable electronics. Due to the hydrogel substrate’s stretchability and tissue-like qualities, it has recently been proposed as a new generation substrate of piezoelectric devices [34]. Therefore, it is necessary for a deep understanding of recent progress and development of hydrogel-based piezoelectric devices. In this paper, we will discuss the advancements that have been made and the advantages that hydrogel-based piezoelectric electronics have over traditional devices in terms of wearable devices and biomedical electronics. When the benefits of hydrogel and piezoelectric materials are combined, the result is a significant performance gain that may be applied to a greater application area for wearable electronics and biological monitoring, including pressure/strain sensors, flow sensors, energy harvesters, wound healing, and ultrasound imaging and stimulation (Figure 1). The role of hydrogel and piezoelectric materials are discussed and summarized, respectively. The future perspective and suggestions are also provided.

## 2. Fabrication and Synthesis of Hydrogel-Based Piezoelectric Composites

As biofunctional materials, hydrogels have a high water content and hydrophilic networks that enable them to retain their structure in water without disintegrating [35]. This polymer with cross-links is the product of the reaction of one or more monomers. Hydrogels can be utilized for drug delivery, sensors, wound healing, energy harvester, soft robotics, and artificial muscles by modifying their material composition and chemical structure [36]. Hydrogels are typically made from cross-linkers, initiators, and monomers. Depending on the hydrogel material, they can also be categorized as natural polymer hydrogels, synthetic polymer hydrogels, or a combination of the two. To suit the requirements of various applications, hydrogels can be synthesized using a number of preparation techniques.

In strain/stress sensor applications, the hydrogel must be able to endure the external mechanical force, which necessitates the hydrogel’s mechanical properties. Traditional piezoresistive strain sensors can detect static loads effectively, but this sensor’s limited sensitivity and poor mechanical characteristics must be addressed at this stage [37,38]. The performance of conductive hydrogels can be substantially influenced by hydrogel materials and conductive network materials. The performance of conductive hydrogels can be substantially influenced by hydrogel materials and conductive network materials. The hydrogel substrate materials can be chosen from a variety of natural and synthetic polymers. Different combinations can be used based on the strain/stress sensor’s needed sensitivity and mechanical qualities [39]. In energy harvester applications, biomaterials, such as silk protein, alginate, cellulose, and chitosan, have attracted much attention. In addition, the use of freeze-resistant hydrogels while increasing the energy output of energy harvesters is also a factor to be considered in the synthesis of hydrogels at this stage.

For example, the typical polyvinylidene fluoride (PVDF) with hydrogel composites are typically fabricated for this application. Polyacrylonitrile–polyvinylidene fluoride (PAN-PVDF) hydrogel can be used as an artificial skin to provide tensile and pressure response. The PVDF was dissolved by adding organic solvent, dimethyl sulfoxide (DMSO) solution; then acrylonitrile (AN), sodium p-styrenesulfonate (NaSS), and methylene-bis-acrylamide (MBA) cross-linker were added sequentially and vortexed to dissolve. Finally, a thermal initiator was added and the mixture was poured into polydimethylsiloxane (PDMS) molds and placed at 60 °C for 1 h to cross-link. The obtained hydrogels were placed in water for 5 days, with water changes every 12 h for impurity removal [40]. In addition, some conductive materials, including carbon materials and conducting polymers, can also be used to improve the performance. A poly(3,4-ethylenedioxythiophene) polystyrene sulfonate (PEDOT: PSS) solution was added to chitosan quaternary ammonium salt, and NaOH was added to keep the pH of the solution at 10. The mixture was stirred until it was completely dissolved. After that, epichlorohydrin was added as a cross-linking agent, and a 10% HCl solution was used to bring the pH to 7. The dialysate was made when cellulose powder was added last and mixed well. The solution was then poured into a mold and concentrated in a vacuum drying oven at 70 °C to make a conductive hydrogel [41]. In another work using a similar method, multi-walled carbon nanotubes (MWCNT) and chitosan were added to 15 mL of acetic acid, then mixed with lauryl methacrylate to obtain the hydrogel by adding N,N,N′,N′-tetramethylethylenediamine [42]. When hydrogel is employed in flow sensors, it is typically created as a cupula structure that can cover nanofibers and expand the sensor’s surface area in contact with the water flow, hence reducing friction [43]. Under the influence of water flow, the hydrogel can deform mechanically and transmit stresses to the substrate to generate output signals, increasing the sensitivity of the sensor. Hyaluronic acid tyramine (HA-Tyr), hyaluronic acid monoacrylate (HA-MA), poly(ethylene glycol) diacrylate (PEG-DA), and other hydrogels are now used in flow sensors. Another justification for using hydrogel materials as flow sensors is their hydrophilic nature [44,45].

Hydrogels used for wound treatment must be biocompatible and breathable, and certain in vivo implanted hydrogels must also possess biodegradable properties to prevent additional wound-site damage [46]. Additionally, hydrogels utilized for skin wound healing must be self-adhesive and simple to peel. In addition, silver nanowires and antibiotics are utilized in the manufacture of hydrogels to prevent bacterial development. Hydrogels composed of piezoelectric materials, such as PVDF and BaTiO_3_, can also imitate bioelectricity to offer electrical stimulation and facilitate wound healing. For example, poly(vinyl alcohol) (PVA)/PVDF hydrogel can be prepared in a simple two-step process when used to promote osteochondral repair. First, PVA and PVDF are added to a solvent of DMSO to homogenize by stirring at high temperature. After that, the prepared solution is poured into a mold, frozen, and defrosted for three cycles before alcohol dehydration. In another work, gellan and PVA were dissolved in deionized water in a 1:1 ratio and heated to 80 °C and stirred for 3 h. Then PVDF–chitosan–gelatin nanofiber was added to the above solution and stirred for 1 h. The mixed solution was dissolved by magnetic stirring at 80 °C and poured into Petri dishes and stored overnight at room temperature. Finally, the cross-linked films were obtained using the freeze–thawing method to achieve the hydrogel [47].

## 3. Hydrogel-Based Piezoelectric Sensors

### 3.1. Pressure/Strain Sensors

The traditional hydrogels investigated for synthetic skin are primarily piezoresistive varieties. Piezoresistive hydrogels function by changing their resistance to external pressure, which allows them to transform external stimuli like strain, pressure, or vibration into electrical impulses. Numerous brand-new piezoresistive hydrogels have been created, and they perform incredibly well as wearable sensors for tracking human activity thanks to their great sensitivity, stability, and wearability. However, in many piezoresistive composites, it is frequently noted that there is an apparent hysteresis effect. Piezoresistive hydrogels cannot function without an external power supply, which could result in large devices and major application limits. Additionally, the majority of currently known piezoresistive hydrogels have low mechanical durability and cannot shield the human body from harm. Therefore, hydrogel-based piezoelectric has appeared as an exciting method that is more suited for artificial skin. The recent studies on hydrogel-based piezoelectric devices for sensing applications are listed in Table 1.

The most selected combination is to use PVDF-based piezoelectric polymers with hydrogels. Fu et al. developed a self-powered PAN-PVDF (10%) energy-harvesting skin for detecting physiological signals such as pulse and hand gestures [40] (Figure 2a). PAN hydrogels are very strong because of intermolecular cyano (CN) dipole interactions, and the dipole interactions between PVDF and CN groups result in a β-phase crystalline content of 91.3%. This energy harvesting can also achieve a 30 mv output voltage and an output current of 2.8 μA with a fast response time (31 ms), which can respond to 30 mg of rice released from a height of 10 cm. This response time is much faster than that of piezoresistive sensors (138–410 ms) [48,49]. PAN-PVDF energy harvesting can also detect arterial pulse signals and detect vocal cord vibrations to interpret what people are saying, although only simple words can be recognized at this stage; this type of energy harvesting still presents great potential for application. In another work, the chitosan quaternary ammonium salt (CHACC)/PEDOT: PSS/poly(vinylidene fluoride-co-trifluoroethylene) (PVDF-TrFE) hydrogel was created using a one-pot thermoforming and solution exchange technique [39] (Figure 2b). The hydrogel-based strain sensor has good mechanical qualities (elongation at break up to 293%), a very high sensitivity or gauge factor (GF: 19.3), a quick response (response time: 63.2 ms), and a wide frequency range (response frequency: 5–25 Hz). Adding PVDF-TrFE enhances the hydrogel’s ability to sense strain, which results in a wider range of responses to both dynamic and static stimuli and a bigger shift in the relative resistance to stress. This has numerous uses, including tracking human motion, spotting minute motions, figuring out object outlines, and using a hydrogel-based array sensor. The construction of composite hydrogels based on piezoelectric and piezoresistive sensing with applications for wearable sensors is also discussed.

To improve the performance, some researchers used inorganic piezoelectric fillers. Expanding the applications of triboelectric nanogenerators (TENGs), particularly for flexible and wearable bioelectronics, is dependent on TENGs’ output performance. Wang et al. proposed an asymmetrical polyacrylamide and BaTiO_3_ (PAM/BTO) composite film, which is composed of BTO nanocubes and PAM hydrogel [37]. The performance of the sensor can be tailored by varying the quantity and distribution of BTO. The stretchable hydrogel electrode could withstand being stretched more than eight times. As pressure sensors, the device exhibited high sensitivity to discriminate a spectrum of forces (0.25–6 N) at the low frequency by altering the content and distribution position of BTO in the unsymmetrical hydrogel film. The improved piezoelectric sensors are employed as multimodal biomechanical sensors for very sensitive detection of human body motions, pressure, and curvature by merging piezoresistive, piezoelectric, and triboelectric effects.

### 3.2. Flow Sensors

Piezoelectric flow sensors are self-powered, and as a result, they do not need to be connected to an external power supply to acquire the sensor output [19]. In the past, the development of MEMS piezoelectric flow sensors mostly utilized PVDF and inorganic fillers as its two primary materials of choice. In this section, the differences between these two kinds of MEMS flow sensors will be discussed.

PVDF possesses outstanding mechanical and piezoelectric qualities, nonlinear optical properties, and flexibility [57,58]. In the past 10 years, there has been a notable increase in interest in the use of PVDF nanofiber in a variety of applications, from energy harvesting to sensors [59,60,61]. PVDF has paired the piezoelectric properties of PVDF with the flexibility of the PDMS microfluidic channel to create a gas-flow sensing device that is flexible and has an exciting possibility for application in microfuel cells or biosensors. Asadnia et al. reported an artificial fiber bundle prepared using PVDF nanofibers and covered with cupula-structured hyaluronic acid-methacrylic anhydride (HA-MA) hydrogel, which deflects and stretches the PVDF fibers under the driving of a water flow to generate an electrical charge [62] (Figure 3a). PVDF nanofibers are encapsulated in the tip of a gradated height PDMS micropillar array to increase sensitivity, which can obtain a sensitivity of 300 mV/(m/s) and a detection limit of 8 μm/s when used as a sensor.

Unlike Asadnia’s gradient height micropillar array, Bora et al. proposed a structure that encapsulated copper pillars and lead zirconate titanate (PZT) piezoelectric membrane substrate in an HA-MA hydrogel cupula [43]. A hydrogel cupula was attached to the stiff cylindrical Cu pillar, which was put atop a PZT piezoelectric sensor membrane (Figure 3b). In comparison to the bare construction, the optimized HA-MA artificial cupula enhanced the mechanical properties of the sensor, increasing its sensitivity and resilience. In their other work, when water flow striking the cupula structure causes the copper pillar to bend, the piezoelectric film can be displaced, thus converting the mechanical signal into an electrical signal. The addition of the cupula structure increases the sensor output by approximately 2.1 times compared to the copper column/PVDF nanofiber structure alone. At water velocity of 0.625 m/s and vibration frequency of 205 Hz, this device can produce an output voltage of approximately 0.1 mV, which can also be used to guide the design of flow sensors for robotic applications [63]. The same group also proposed a carbon nanotube bundle/PVDF membrane encapsulated in hyaluronic acid–tyramine (HA-Tyr) hydrogel cupula structure that shows a high sensitivity to water flow and air flow. This energy harvester is capable of producing an output voltage of 100 mV for a weak airflow (10 mm/s) and a sensitivity of 6.5 mV/μL for different volumes of falling water when the device is submerged in 2 cm deep water. This three dimensional structure supported by carbon nanotubes simplifies the fabrication steps of previous microelectromechanical system sensors and improves the sensitivity of PVDF-nanofiber-based membranes without the need for an external power supply [64].

In some work, the PVDF and inorganic fillers were used together to improve the performance. Ma et al. developed a flow sensor by using a BTO/P(VDF-TrFE) mat and poly (ethylene glycol) diacrylate (PEG-DA) hydrogel, which is an artificial cupula. Due to the improved piezoelectricity (34 pm/V) and the use of a cantilever structure, the sensor has a detection limit of 0.42 mm/s, which is a great improvement compared to the H-SALL sensor (5 mm/s) of Bora’s results [64]. A constriction structure was added into the canal to increase the flow speed within the canal, thus improving the sensitivity of this energy harvester, which has a detection limit of 5.89 Pa/m for hydrodynamic pressure gradients. This work demonstrates the great potential of this flow sensor. To enable the flow sensor to detect the direction of water flow, Guo et al. proposed a stepped micropillar structure sensor in the height range of 40–1000 µm with a sensitivity of 1.18 mV/(mm/s). Since the output signal of the microcolumn structure in the inhibited direction (3 mV) is smaller than that of the excitatory direction (6 mV), it can be used for direction identification and can also detect water velocities down to 2 mm/s. This work provides a new method for water flow direction detection [45].

In summary, the soft hydrogel cupula is given mechanical strength by the nanofibers, increasing its endurance for underwater sensing applications. Comparing the artificial cupula sensors to the bare hair cell sensors, the artificial cupula sensors showed a significant increase in sensor sensitivity. The need to waterproof the sensing components and sensor contacts makes measuring flow velocity and flow direction in water a more difficult process. Future development of nature-inspired micro- and nano-flow sensors will be guided by the biomimetic approach and unique manufacturing techniques outlined in this research.

## 4. Hydrogel-Based Piezoelectric Energy Harvesters

The use of wearable and implantable medical devices, which can transmit the acquired biosignals to the user in real time, has greatly improved people’s life quality [3,65]. These devices can provide massive data support for health monitoring [65,66,67,68], sports training [69,70,71], and disease screening [72,73,74]. Although the usage of batteries for power supply can increase the portability of the device, the rigid structure of batteries limits the device’s flexibility, and the battery’s lifespan can impact the user experience. Changing batteries frequently is not only inconvenient and uncomfortable for devices like cardiac pacemakers and implantable defibrillators, but it can also easily cause secondary injury to the patient [75]. Therefore, it is vital to develop new and sustainable energy alternatives. The skin surface generates a lot of energy during regular activities that can be converted by energy-harvesting devices into electricity for powering wearable devices [76]. Using the piezoelectric effect of materials, energy harvesting is able to provide output voltages from microwatt to milliwatt to power various types of wearable devices [77]. Unlike batteries, energy harvesting devices are self-powered and rarely require frequent replacement. Piezoelectric generators, which convert the mechanical energy of the human body into electricity, have been widely studied and applied to the detection of variety of human signals. Recently, many inorganic (piezoelectric materials) and organic (soft matrix) composites have been suitable in piezoelectric-based flexible electronics. Various polymers, such as polyvinylidene fluoride (PVDF), polyvinyl chloride (PVC), silicon elastomers polydimethylsiloxane (PDMS), Ecoflex, and conducting polymers (PANI, PPy, etc.) [27,60], have been used as matrix or substrates. Compared to these soft materials, hydrogels exhibited increased the permeability, comfortability, and biocompatibility of wearable devices, thus providing more possibilities for human-signal monitoring. The recent studies on hydrogel-based piezoelectric devices for energy-harvesting applications are listed in Table 2.

Because of the similar design structures and working principle, most of hydrogel-based piezoelectric sensors can also play the role of energy harvesting [83]. As we discussed in Section 3.1, Fu et al. proposed the PAN-PVDF sensor, which is also capable of generating an electrical signal output (~30 mV and ~2.8 µA) with a rapid response (~31 ms) because of the stress-induced poling effect [40]. After this work, some PVDF-based energy-harvesting devices have been also proposed. Jia et al. developed a stretchable and biocompatible energy harvester for monitoring joint motion during basketball training by encapsulating PVDF in PAAM-LiCl hydrogel. This sensor was able to generate an output voltage of 4.1 V and 1.99 V at bending angles of 65° and 138°, respectively, and is not affected by sweat or rising body temperatures. When working as an energy harvester, it is also capable of charging a 2.2 μF capacitor to 1.7 V and a 4.7 μF capacitor for 30 s to drive a portable calculator, showing potential as a next-generation motion detector [81] (Figure 4a). This experimental group also used PVDF and PAAM-LiCl hydrogels to prepare nanogenerators to monitor and analyze the movements of divers. This sensor can combine the advantages of PENG and TENG for the accurate monitoring of complex movements, with output voltages of 2.45 V and 1.25 V before and after entering the water, respectively. By monitoring the athletes’ wrist movement information, it can guide the athletes’ movement into the water and help reduce the splash generated by the athletes when entering the water [82] (Figure 4b).

Compared to PVDF film, the ceramic filler always showed enhanced piezoelectric performance. Zhang et al. proposed a BaTiO_3_/bacterial cellulose hydrogel nanogenerator that can generate a 14 V open-circuit voltage and 190 nA/cm^2^ current density. Directly connected LCD devices can be illuminated without rectifier units when the nanogenerator is bent and released by the finger, and the peak voltage reaches 1.5 V under 1 Hz deformation. Stable performance over 3000 cycles also indicates that the device has strong mechanical properties [78]. By encapsulating BaTiO_3_/nano tubes in bacterial cellulose hydrogel as a piezoelectric layer and using nitrocellulose nanofibril as a triboelectric layer, a tribo/piezoelectric nenogenerator for energy harvesting has been proposed. This sandwich-structure energy harvesting has an output voltage of 18 V and an output current density of 1.6 µA/cm^2^ where the short-circuit current density and the open-circuit voltage of the piezoelectric part are 220 nA/cm^2^ and 22 V, respectively. After rectification, three LED bulbs can be directly driven by this device and the 1 μF capacitor can be charged to 2.5 V in 80 s. When the device is used as a pressure sensor, the sensitivity can reach 8.276 V·cm^2^/N [79]. Gao et al. proposed an energy-harvesting device with piezoresistive and piezoelectric properties using PVDF-TrFE/ZnO and PEDOT: PSS hydrogel to identify the profile of the grasped object. This device achieves an average identification rate of 84% for eight different shapes and sizes of objects and can produce an output voltage of 1.45 V at a load of 2 N. The combination of piezoelectric and piezoresistive sensors not only enables the robot to detect dynamic and static forces simultaneously, but also significantly increases the precision of recognizing object contours, which provides a new idea for unmanned sorting of intelligent robots [41].

In addition to chemical hydrogels, biomaterial-based hydrogels have also been used in bio-inspired technologies such as photonics, electronic skin, and soft robotics. These technologies have been inspired by biological systems [84]. Silk protein, which has been designed in such a way as to induce the hydrogel form, is one type of biomaterial that is used in tissue engineering, drug administration, and different sensing applications. Although silk protein demonstrated piezoelectric capabilities, the outputs of the previously reported silk films and nanofibers were much lower in contrast to those of standard inorganic piezoelectric devices. To enhance the performance, a ZnO nanorod was added because it is a biocompatible, cost-effective, and physically/chemically stable material with a large piezoelectric constant. Gogurla et al. reported energy-harvesting skin by using ZnO nanorods and biocompatible silk hydrogels, which can generate both triboelectric and piezoelectric effects and detect biomechanical motion [80] (Figure 5). The idea behind the composites is to use a silk hydrogel with ZnO nanorods incorporated into it. This material closely resembles soft skin or biological tissue and has the ability to produce electrical energy through hand-stimulation-induced piezoelectricity (in the Zn nanorods and silk crystalline regions) and triboelectricity (on the silk surface) at the area where the EG-skin is conformally attached. This energy harvester is capable of generating the power of ~1 mW/cm^2^ for a load resistance of 70 MΩ to drive a stopwatch and an LED after charging the capacitor for 360 and 480 s, respectively. In addition, this electronic skin can be used as a sensor to monitor muscle strain, biomechanical movement, and breathing activity. The absence of allergy after 12 h of adhesion to the skin indicates that this e-skin is biocompatible. The silk protein hydrogel is able to degrade completely after one hour when catalyzed by protease. The EG-skin-generated power was sufficient to switch on the oximeter to monitor human health conditions, indicating that the EG-skin was applicable as a power source for low-power-consuming biomedical devices.

## 5. Hydrogel-Based Piezoelectric Devices for Wound Healing

Skin damage can be caused by a variety of factors, including surgery, accidental injury, and disease. The skin serves as the body’s natural barrier against bacterial attack, and when it is damaged, the body initiates a series of damage-repair and protection mechanisms. The skin first goes through an inflammatory phase to remove external cells or impurities, followed by a tissue formation phase in which structures such as epidermal cells and tissues such as blood vessels are gradually repaired. The final remodeling phase attempts to return the wound as close to its pre-injury state as possible [85]. Wound management is critical for tissue regeneration and scar reduction. Gauze has traditionally been widely used as an external barrier on wounds, and as medical technology has advanced, effective approaches to accelerating wound healing have been sought [86]. Endogenous electric fields (EFs) are targeted signals for cell migration and tissue repair when skin epithelial tissue is disrupted. Since 1843, when Emil Du Bois-Reymond first measured electrical activity associated with wound and nerve excitation, the study of electrical signals in the human body has received considerable attention [87].

Due to the electrotaxis nature of cells, the cells inside the wound will migrate directionally under the guidance of EF (electric field) [88]. As shown in Figure 6, the rate of wound healing, however, gradually slows down as the bioelectricity released by the surrounding tissues decreases throughout the gradual wound-healing process [89]. Recent studies have shown that the use of applied electrical stimulation to treat wounds can promote wound healing and that EFs play a crucial role in wound healing [88]. The main types of currents applied for wound healing at this stage are direct current lasting one second or longer, single-phase pulsed current, and biphasic pulsed current [87]. In addition, wound tissue heals more easily in a moist environment, which provides a theoretical basis for the use of hydrogel as a wound-healing patch matrix [85]. The use of a combination of piezoelectric materials on a hydrogel matrix to provide electrical stimulation to the wound is a viable option for accelerating wound healing.

Numerous flexible piezoelectric skin patches for wound healing have been developed in recent years. Flexible piezoelectric materials can efficiently address the issue of bulky equipment in conventional electrical stimulation therapy and enable real-time treatment because they are self-generating and lightweight. Additionally, by applying the patch directly to the wound surface and generating a moist healing environment, the biocompatible hydrogel used as the substrate facilitates quick wound healing. To give external electrical stimulation, skin patches that utilize piezoelectric materials on a hydrogel matrix have drawn a lot of attention. The recent studies on hydrogel-based piezoelectric devices for wound healing are listed in Table 3.

Piezoelectric skin patches made with a PDMS matrix can effectively facilitate re-epithelialization and tissue regeneration, which promotes the healing of dorsal skin wounds (1.8 × 1.8 cm^2^) in mice within 10 days. However, the poor adhesion and permeability of PDMS and the large difference with skin modulus limit its further clinical application [95]. To address this issue, Du et al. demonstrated a bio-inspired patch with self-adhesive and natural wound EF using a chemical hydrogel made of polydopamine–polyacrylamide (PDA-PAAm) and piezoelectric stimulation generated by electrospun poly (PVDF) nanofibers. The improved adhesive properties of the patch are mainly attributed to the catechol group in the hydrogel, and the excellent permeability of PDA-PAAm (91.4 μLh^−1^) also reduces the risk of excessive local hydration while maintaining proper wound moistness. Mice in vivo experiments show that with movement, the patch can produce an average voltage of 0.1–0.5 V, which can stimulate complete healing of a 5 mm diameter circular wound on the back by day 10 [86] (Figure 6a). Similarly, Sharma et al. developed a skin patch using PVDF as a piezoelectric layer to generate electrical pulses and carbonized PDA (CPDA) as a hydrogel conductive layer for the treatment of diabetic foot ulcer (DFU). It is noteworthy that CPDA has higher strength than conductive hydrogel. In addition, the CPDA hydrogel can self-heal and resist stretching after being cut and kept in contact for 8 min. This skin patch has been shown to promote wound epithelialization and vascularization, thereby accelerating wound healing [90] (Figure 6b). Chen et al. also used a mussel-inspired PDA hydrogel as a substrate to produce CM@DA films by coating chitosan films with polydopamine. Using the photothermal effect of PDA (generating heat under near-infrared irradiation) and the piezoelectric effect of chitosan, this group reported the first protocol for electrical stimulation and heat to accelerate wound healing. In vivo experiments on the backs of male mice demonstrate that the CM@DA skin patch can rapidly warm up and stay at 45 °C within 5 min under near-infrared irradiation (NIR). Electric stimulation and heat can synergistically promote collagen deposition and Hsp90 and HIF-1α expression, which in turn promote wound healing. Experiments have also shown that both PDA coating and NIR irradiation facilitate the enhancement of chitosan piezoelectric properties, though the precise mechanism behind this enhancement is unknown and needs to be studied further [93]. In addition, Mohseni et al. designed a substrate that encapsulated PVDF–chitosan–gelatin nanofibers and mesoporous silica nanoparticles (MCM41) into gellan–PVA hydrogel. It is worth noting that MCM41 nanoparticles are used to increase the voltage output and antibacterial activity. Although these two skin patches have not been tested in vivo, they still offer a new attempt at wound-dressing applications [47].

Instead of using PVDF as a piezoelectric material, other piezoelectric materials have also been adopted to provide stimulation. Goonoo et al. introduced a 20/80 PDX/PHBV mat to prevent scarring in tissue regeneration. The use of electrospinning allows the synthesis of 20/80 scaffolds with a good hydrophilic/hydrophobic balance, and it can also reduce fibroblast-mediated contraction deformation during the healing process of circular wounds with a diameter of 6 mm on the backs of mice, thus reducing scarring after three weeks. The good mechanical and piezoelectric properties of this scaffold can effectively improve signaling channels and promote angiogenesis. It also enhances the growth of keratinocytes and reduces the inflammatory response [92]. Bai et al. used ZnO nanorods as a piezocatalytic and nanozyme catalytic layer, which was encapsulated in a graphdiyne nanosheet to provide a piezoelectric response. PMDS was used as a matrix to form a skin patch for promoting wound healing. The use of the enzyme hydrogen peroxide catalase, which promotes the decomposition of hydrogen peroxide to produce oxygen, not only prevents anaerobic bacteria from proliferating in the wound, but also accelerates the healing of the wound tissue. It is worth noting that this novel skin patch was able to heal a 7 mm diameter wound on the back of a mouse in 9 days [94]. Liang et al. also innovated in the preparation technique by producing the first PVDF/sodium alginate (SA) scaffold with 0.5% ZnO nanoparticles using a 3D printing method. A 3D-printed preparation of skin patches has not been reported previously, mainly because of the difficulty of mixing hydrophobic piezoelectric materials with hydrophilic hydrogels. To solve this problem, they tried different PVDF and SA concentrations after enhancing the hydrophilicity of PVDF using ZnO. The results showed that 8% PVDF and 6% SA bioink would be able to maintain a fine structure and uniform porosity. What is more, this scaffold was able to achieve horizontal triboelectric and vertical piezoelectric effects, which allowed 1 × 1 cm wounds on the backs of the SD rats to heal completely by day 14. This skin patch can generate frictional electricity at the same time in the swollen state, in which the output current can reach ±1.29 µA [89].

Electrical stimulation treatments not only accelerate the healing rate of skin wounds, but also promote the repair of nerve tissue and cartilage. Tan et al. used PVDF material to generate electrical stimulation and loaded CXCL12 and G-Exos drugs in methacrylate gelatin gels (rGel) to collectively promote nerve growth at skin wound sites within 23 days. It is worth noting that this design enables the patch to bend 0.1% under an external force of 0.1 N, resulting in a voltage of 35 mV. In total, 42% and 63.7% of CXCL12 and G-Exos were able to be released within 48 h, respectively [96]. In the direction of cartilage therapy, Wu et al. proposed a PVDF/PVA bilayer structure incorporating 0.3% silver nanowires. In vitro experiments showed that this material was able to generate 0.32 V. In vivo experiments showed that rabbit osteochondral bone was able to recover after 12 weeks, which indicates that piezoelectric hydrogel patches have promising potential in tissue engineering [46].

## 6. Hydrogel-Based Piezoelectric Ultrasonic Devices

### 6.1. Ultrasound Stimulation

Ultrasound is widely used in clinical medical diagnosis and treatment because of its better tissue penetration ability and non-invasive nature [97]. The use of an ultrasonic-stimulation piezoelectric hydrogel patch also has promising application prospects. The main working principle in the medical field is that ultrasound stimulates the deformation of piezoelectric materials and thus generates electrical signals, which can be used to promote tissue regeneration and promote drug release for cancer treatment [98,99]. Hydrogel patches with self-powered ultrasonic transducers can also be adhered to human organs for ultrasound detection, and in the in vitro phase, the ultrasound response can be tested by using ultrasound to stimulate transducers [100,101]. In the field of water purification, piezoelectric materials can be driven by water waves to produce deformation, which in turn generates electrical energy to accelerate the evaporation of water from hydrogels [102]. Ultrasound stimulation of biological implants can supply power to the implant in all directions, thus avoiding the use of batteries and facilitating a more flexible design of the device [103,104]. The recent study on hydrogel-based piezoelectric devices for ultrasound stimulation are listed in Table 4.

Using BaTiO_3_@alginate/Pluronic bioink, Paci et al. developed a 3D bioprinting hydrogel to investigate the effect of piezoelectric effects on early myogenic differentiation using ultrasonic stimulation. Alginate is mainly used to promote cell proliferation and adhesion because of its excellent biocompatibility and gel synthesis ability. Examples of 3D bioprinting using alginate bioinks for bone–muscle constructs have been widely reported [106,107,108,109]. Experiments showed that there are no adverse effects on the biological activity of cells at a nanoparticle concentration of 250 µg/mL. Overexpression of MYOD1, MYOG, and MYH2 genes occurred when samples were stimulated by ultrasound for three continuous days during differentiation, which indicates ultrasonically stimulated barium titanate nanoparticles can be utilized to promote myoblast differentiation in 3D bioprinted configurations [105]. Zhu et al. reported an energy harvester for tumor (4T1 breast cancer) therapy by embedding tetragonal BaTiO_3_ nanoparticles in a thermosensitive hydrogel (chitosan hydrogel), which can generate toxic hydroxyl and superoxide radicals under the stimulation of ultrasound. The results of in vivo and in vitro experiments show that this treatment protocol is biocompatible, and a tumor can be completely eradicated within five days after three treatments without recurrence. This piezoelectric catalytic treatment of tumors through mechanical energy stimulation offers a promising strategy for non-invasive therapy [98] (Figure 7a).

In the biomedical direction, Hermenegildo et al. introduced a scaffold incorporating CoFe_2_O_4_/PVDF into a methacrylated gellan gum (GGMA) hydrogel for stimulating tissue regeneration. Under the stimulation of an applied magnetic field, the deformation of the magnetically sensitive material stimulates the piezoelectric material and generates an electrical signal. This porous scaffold has a piezoelectric response of 22 pC/N and good biocompatibility, providing a new direction for stimulating in vitro cell regeneration by simulating a bioelectric microenvironment [99]. By using PVDF piezoelectric film and poly (methacrylic acid-co-acrylamide) hydrogel, Islam et al. designed a bilayer transducer for detecting pH changes in bladders to provide information for disease prevention. To verify the feasibility of this design, in the actual experiments, the bending angles of hydrogels were 41.3° and 125.1° when the pH was 4 and 12, respectively. In addition, the sensitivity of 0.006 mV/cm^2^/pH and 0.017 mV/cm^2^/pH was achieved by using a 5 Hz, 225 mV pulse signal to drive the third- and fourth-order bilayer transducers, respectively. This design provides a new approach to wireless chemical sensing [100].

Based on the need for implantable passive sensors for strain detection in human organs, which was able to achieve a maximum of 260% stretchability, ultrasonic images of the hydrogel can be easily created by scanning the hydrogel covered with porcine skin and the size of the deformation of the hydrogel can be quantified and analyzed. When this hydrogel patch is attached to the balloon, it can change cyclically with the contraction and expansion of the balloon and will not detach from the surface. The mechanical properties can be adjusted by modifying the amount of ZnO to meet the needs of different biological tissue detection [101] (Figure 7b). To improve the water purification efficiency, Meng et al. proposed a method that uses water waves to drive the PVDF-TrFE piezoelectric layer to generate electrical energy and thus activate the water in the PVA hydrogel, which provides a low enthalpy of evaporation (1.077 kJ/g) and a high vapor-production rate (2.702 kg/m^2^ h). The graphene oxide in the hydrogel can also absorb sunlight for photothermal conversion after reduction, and these two synergistic effects provide a prototype for water evaporation [102].

### 6.2. Hydrogel Assisted Ultrasound Imaging

Ultrasound is frequently utilized in the medical field to view the interior of the body without radiation and damage to human tissue [110,111,112]. Because ultrasound has trouble traveling through the air because of acoustic impedance, the ultrasound transducer used to interact with the human body requires a medium called “ultrasound gel” to minimize the side effects that occur when moving the ultrasound transducer on the skin and removing the air to avoid the noise occurring during the imaging [113]. A gel is a thick, jelly-like substance with properties ranging from flimsy and soft to firm tissues. Gels are described as a very weakly cross-linked assembly that exhibits no stream in the steady state. Gels have a three-dimensional cross-linked arrangement inside the fluid, which makes them behave like solids even though they are largely liquid by weight. The cross-linking occurring within the liquid is what gives a gel its structure or hardness. A gel is simply a dispersion of liquid-phase (liquid is the discontinuous phase) and solid-phase (solid is the continuous phase) particles within a solid. Although the gel is sometimes semi-solid and shaped into disks, it may nevertheless transmit ultrasonic waves with ease [114]. The purpose of ultrasound gel is to enhance the transmission of ultrasound waves, which raises the idea of using hydrogels as the coupling media for ultrasounds. Because the hydrogels are mainly structured by the water, which has the features of low impedance that perfectly fit the purpose of the ultrasound gel. Networks of cross-linked and hydrophilic polymers that swell in water make up hydrogels. The advantages of hydrogels include their relatively inexpensive material costs, ability to assume solid shapes, and simplicity of mounting to an ultrasound transducer. They are a desirable option as single-use acoustic couplers for high-intensity focused ultrasound devices because of all these benefits. Very recently, Wang et al. described a wearable ultrasound imaging device that used a rigid piezoelectric probe array bonded to the skin with an acoustically transparent, soft, anti-dehydrating, and bio-adhesive hydrogel–elastomer hybrid [115]. The array was connected to a commercially accessible ultrasound platform, allowing for continuous ultrasound pictures of the carotid artery, lung, and abdomen. In vivo testing demonstrated that the device could be worn comfortably for 48 h.

## 7. Conclusions and Outlook

The most attractive choices for future wearable technology and intelligent systems are hydrogel-based functional devices because of their combination of high sensitivity, outstanding conductivity, and skin-like mechanical performance. The development of hydrogel-based piezoelectric devices, including design concepts and various applications, are summarized in this review. Clearly, hydrogel-based piezoelectric devices have been widely used for pressure sensing, strain sensing, flow sensing, energy harvesting, wound healing, and ultrasound stimulation and images.

For strain/pressure-sensing applications, the detection of human motions and physiological signs has so far been made possible using hydrogel-based strain-and-pressure sensors, including hand gestures, body pulse signals, electrocardiograms, and spoken words. Despite the recent successes of ongoing research, hydrogel-based strain-and-pressure sensors are still in their infancy, and pressing and unmet needs persist in the transition from theory to practical implementations. As listed in Table 1, when designing and making hydrogel-based sensors, sensitivity (or GF), working range, and response time, as three key parameters, should be carefully taken into account. Despite the great sensitivity of nanocomposite hydrogels, there are still some current problems for hydrogel-based piezoelectric sensors. First, the level of sensitivity needs to be evaluated in relation to the other qualities. To recognize more human motions, it is necessary to ensure that the detection range is sufficient and that the operational range is expansive. Because of the incompatibility between nanofillers and the hydrogel network, the stretchability of hydrogels is always going to be limited; as a result, the working ranges for nanocomposite hydrogel-based strain-and-pressure sensors still need to be increased. Second, there is still a challenge regarding the compromise that must be made between resistance to fatigue and a broad working range. In particular, when subjected to extreme strain, hydrogels are unable to return to their initial state, which further creates a considerable level of signal hysteresis.

For energy-harvesting applications, hydrogels have greater elasticity, flexibility, fast diffusion, and biocompatibility when compared to other materials, especially a self-healing hydrogel for the effective electrostatic electrode. Hydrogels currently used for energy harvesting already have good adhesion properties while meeting the energy supply needs of low-power appliances. However, the following are the main difficulties with hydrogel-based energy harvesting preparation: (1) it is challenging to choose the best piezoelectric components and hydrogel for a certain energy harvesting; (2) hydrogel preparation can be difficult, and energy harvesting applications are severely hampered if any breakdown happens; (3) it is challenging to produce an exact mix of initiators and cross-linkers in the polymer matrix, which could provide a risk; (4) due to their weak conductivity, commercial intrinsic hydrogels need to be further functionalized—as a result, creating a suitable hydrogel that satisfies energy-harvesting standards is expensive and time-consuming; and (5) hydrogels are not practical for energy harvesting produced additively. This area needs more investigation. Most essential, the components should be connected together in a way that maximizes power output.

For wound-healing applications, the piezoelectric nanogenerators with biocompatible qualities might be integrated into self-powered bioelectrical systems for medical applications and wearable self-powered treatments with a decade’s worth of work on more efficient energy transfer. The research has shown that hydrogel-based piezoelectric devices are a potential biotechnology that can have a big impact on the treatment of wounds because of its low cost, portability, and real-time effect. At this stage of hydrogel research, improvements have been made in terms of increasing catalase activity, stretchability, and permeability. However, using multifunctional hydrogels for wound-healing applications still presents difficulties despite their potential benefits. For upcoming research, besides the piezoelectric effect, creating sophisticated, multifunctional hydrogels with smart medication-release systems and wound monitoring instruments like pathogenic infection detectors could be a promising strategy. An intriguing strategy is to create multifunctional hydrogels with strong mechanical capabilities, self-healing, antioxidant, antibacterial, cell proliferation, and tissue adhesion properties and take advantage of additional biomacromolecules. In the area of self-assisted wound healing, there is still much room for piezoelectric performance enhancement. More accurate clinical situation-based research and development in this area is needed. To give a brighter future for wound healing, the clinical translation potential and marketing of the prepared hydrogels should be taken into consideration.

For wearable and biomedical applications, environmental stability and long-term stability are crucial. In contrast to conventional substrates, hydrogels will eventually lose water in dry conditions and freeze in extremely cold temperatures. In other words, severe environments will cause hydrogel-based sensors to lose their functionality, making them unsuitable for use. Although various approaches to address these issues have been put forth, including solvent switching, surface modification, and encapsulation, they will either affect the mechanical performance of hydrogels or degrade the function of the integrated sensors. Exploring a new generation of hydrogels that can function properly in challenging conditions without losing their intrinsic features is therefore eagerly anticipated. In addition, hydrogels’ potential to retain water and moisturize should be emphasized. Otherwise, the device’s effectiveness could be impacted by the hydrogel’s quick evaporation rate.

As a future perspective, the development of innovative strategies to broaden the use of hydrogel/piezoelectric composites to improve their service life and performance stability is of the utmost importance. The improved material innovation, design and simulation, and system integration are all important steps. First, the novelty of both hydrogel and piezoelectric materials are highly desirable. Improved mechanical performance can be achieved by introducing double cross-linking, nanofillers, ionic/zwitterionic polymers, and reversible physical connections into the hydrogel network. It is essentially suggested to create a network consisting of physically robust and reversible links between piezoelectric fillers and polymeric chains to combine all of the necessary qualities in a hydrogel-based device. With so many physical linkages, the hydrogel is given a self-healing quality, which enables high-efficiency mechanical and electrical property recovery after damage. Second, to better understand hydrogel performance, deformation, and recovery with respect to materials and component structure, enable improved design, and enhance response of hydrogel-based electronics, multiscale computational modeling and simulations should be conducted at different levels, including the atomic, molecular, and macroscopic levels in conjunction with experimental analysis of polymer chemistry, synthesis, and gelation. Third, the system-level consideration is critical to the practical applications. Most activities have been dedicated to improving performance compared to the integration of devices with power supply, signal conditioning, and data acquisition unit. It is necessary to reduce the influence from the system on the performance of hydrogel-based devices.

While hydrogel-based piezoelectrics are still in their infancy and the number of unresolved critical issues should be addressed, their high performance in bio-medical applications (sensing, energy harvesting, stimulation, wound healing, implantable bioelectronics, etc.) has prompted the scientific community to focus more efforts on understanding/revealing the fundamental science and to advance its application. Future advancements in intelligent and multifunctional conductive hydrogels that mirror the operation of these incredible natural systems will change a range of biomedical applications, including health monitoring, and result in numerous enhancements to our quality of life.

## Figures and Tables

**Figure 1 micromachines-14-00167-f001:**
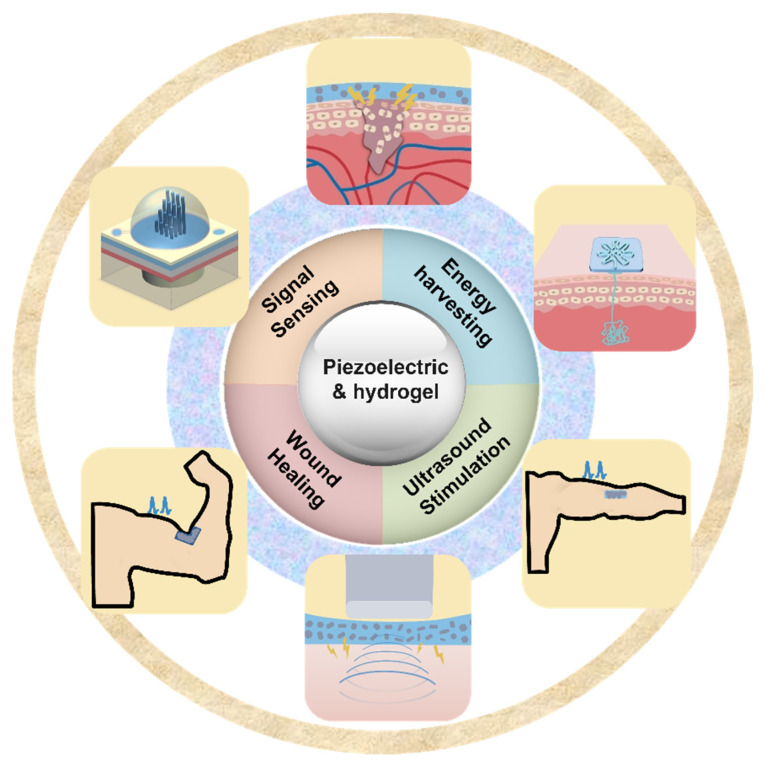
Overview of hydrogel-based piezoelectric devices for biomedical applications, including pressure/strain sensing, flow sensing, energy harvesting, would healing, and ultrasound stimulation/imaging.

**Figure 2 micromachines-14-00167-f002:**
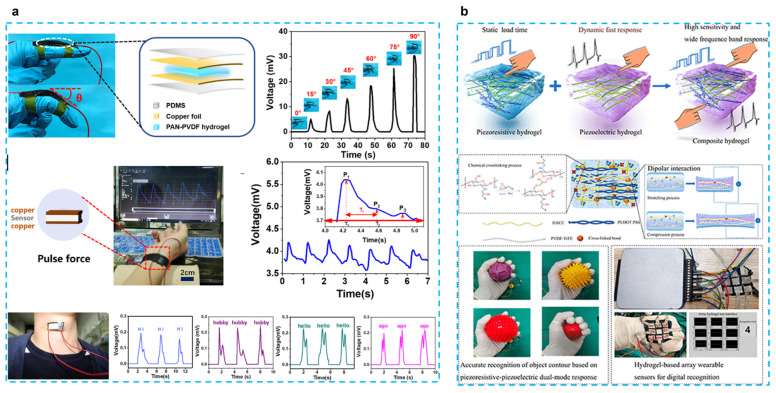
(**a**) A tough and self-powered hydrogel for artificial skin. (Reprinted with permission from Ref. [40], Copyright 2019, American Chemical Society.) (**b**) Enhancing strain-sensing properties of the conductive hydrogel by introducing PVDF-TrFE. (Reprinted with permission from Ref. [39], Copyright 2022, American Chemical Society.)

**Figure 3 micromachines-14-00167-f003:**
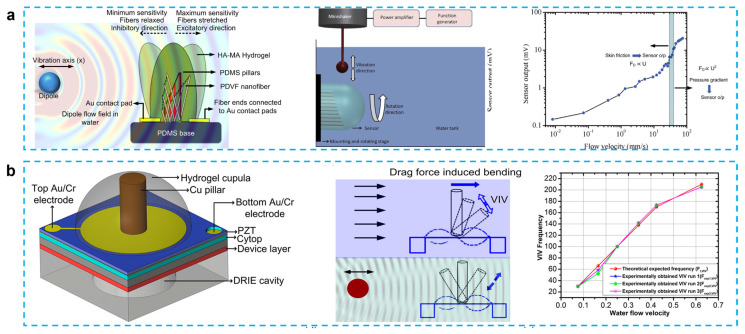
(**a**) From biological cilia to artificial flow sensors: biomimetic soft polymer nanosensors with high sensing performance: a schematic showing the flow sensing mechanism in the presence of an oscillating dipole, schematic showing the basic experimental setup used in all the experiments, and sensitivity and threshold-detection limit of the sensors over a range of velocities. (Reprinted with permission from Ref. [62], Copyright 2016, Springer Nature.) (**b**) Fish-inspired self-powered microelectromechanical flow sensor with biomimetic hydrogel cupula: biomimetic flow sensor with an artificial cupula, schematic showing sensing mechanism of steady-state and oscillatory flow stimuli, steady-state flow sensing using flow-generated VIV. (Reprinted with permission from Ref. [43], Copyright 2017, AIP.)

**Figure 4 micromachines-14-00167-f004:**
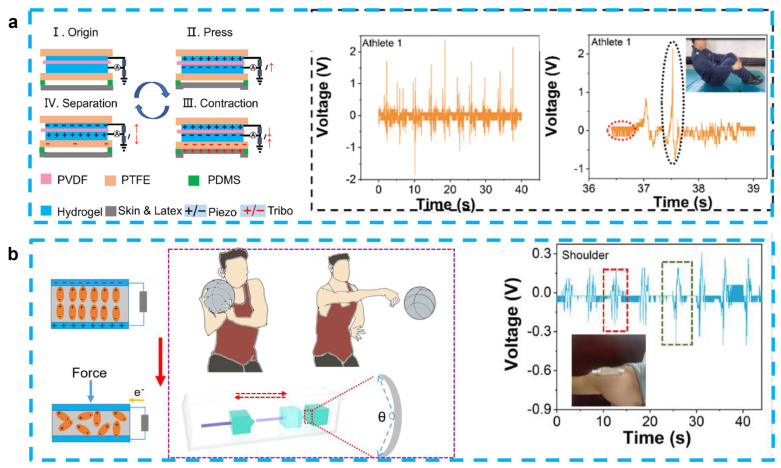
(**a**) A stretchable and self-healing hybrid nano-generator for human motion monitoring: coupling mechanism of TENG and PENG, outputting voltage and details of Athlete 1’s 301C diving motion. (Reprinted with permission from Ref. [81], Copyright 2021, MDPI.) (**b**) A flexible and stretchable self-powered nanogenerator in basketball-passing technology monitoring: working mechanism and scene of TSB-PENG monitoring the body joint motion system, outputting piezoelectric voltage of TSB-PENG on the shoulder that does the bend-and-stretch motion. (Reprinted with permission from Ref. [82], Copyright 2021, MDPI.)

**Figure 5 micromachines-14-00167-f005:**
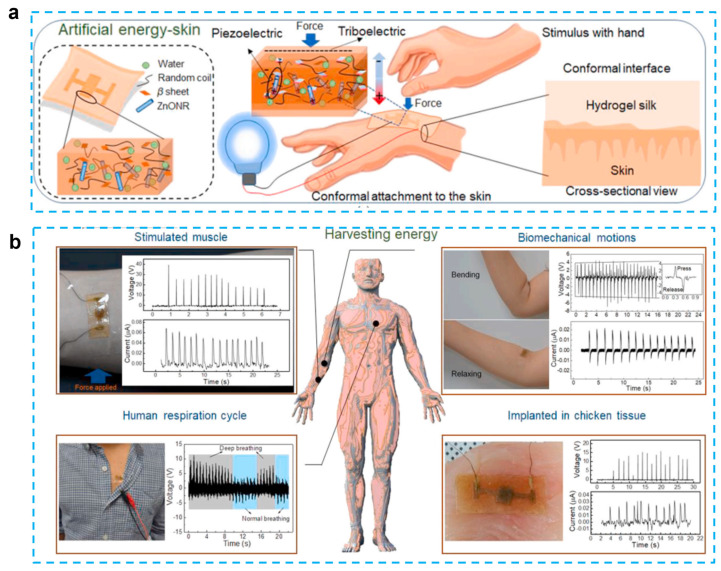
Self-powered artificial skin made of engineered silk protein hydrogel. (**a**) Schematic illustration to show the concept and working principle of artificial EG-skin using silk hydrogel. (**b**) Energy generation from the EG-skin attached on forearm by tapping the hand to stimulate the muscle, elbow by bending and releasing, chest for monitoring respiration cycles, and embedded in chicken breast tissue. (Reprinted with permission from Ref. [80], Copyright 2020, Elsevier Ltd.)

**Figure 6 micromachines-14-00167-f006:**
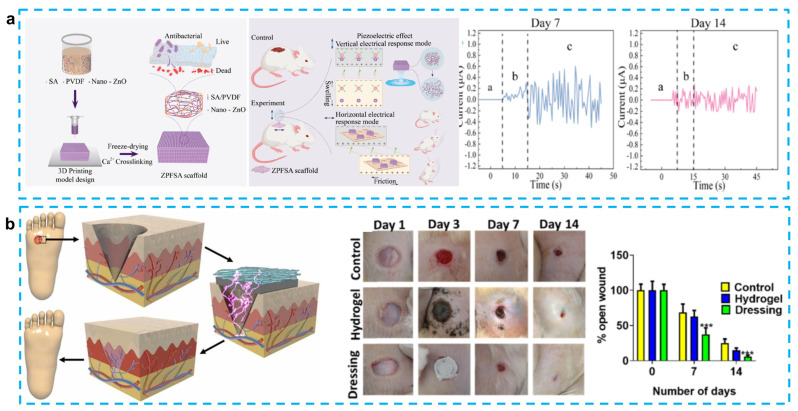
(**a**) 3D-printed piezoelectric wound dressing with dual piezoelectric response models for scar-prevention wound healing. (Reprinted with permission from Ref. [89], Copyright 2022, American Chemical Society.) (**b**) Electrical stimulation induced by a piezo-driven triboelectric nanogenerator and electroactive hydrogel composite accelerating wound repair. (Reprinted with permission from Ref. [90], Copyright 2022, Elsevier Ltd.)

**Figure 7 micromachines-14-00167-f007:**
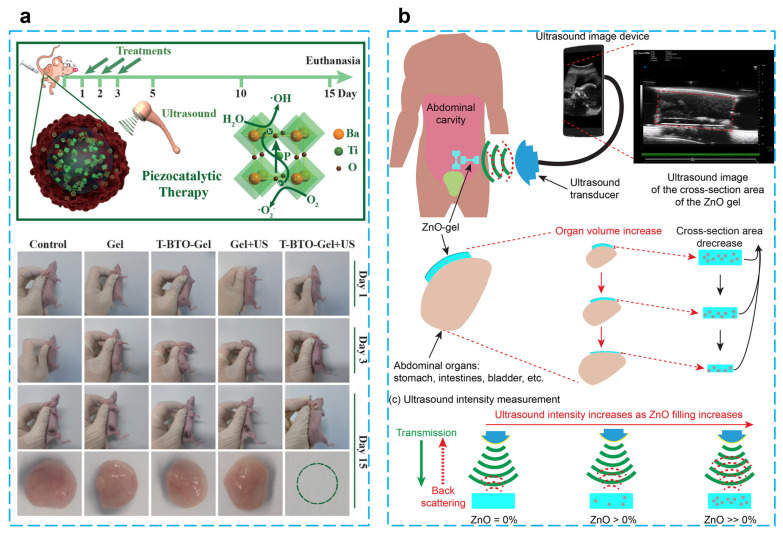
(**a**) Piezocatalytic tumor therapy by ultrasound-triggered and BaTiO_3_-mediated piezoelectricity. (Reprinted with permission from Ref. [98], Copyright 2020, WILEY-VCH Verlag GmbH & Co.) (**b**) A wireless implantable strain-sensing scheme using ultrasound imaging of highly stretchable zinc oxide/poly dimethylacrylamide nanocomposite hydrogel. (Reprinted with permission from Ref. [101], Copyright 2020, American Chemical Society.)

**Table 1 micromachines-14-00167-t001:** Hydrogel-based piezoelectric devices for sensing applications, including critical parameters and applications.

Piezoelectric	Hydrogel	Sensitivity	Operation Range	Response Time	Application	Ref.
PVDF	PAN	-	-	31 ms	Pressure sensor	[40]
PVDF TrFE/ZnO	PEDOT: PSS	-	0.05–0.16 MPa	-	Pressure sensor	[41]
ImClO_4_	Bacterial cellulose	4 mV kPa^−1^	0.2–31.25 kPa	-	Keyboard	[50]
BCE	Tortoise Jelly	1.24 kPa^−1^	0–0.16 kPa	130 ms	Strain sensor	[51]
BaTiO_3_	PAM	0.25–6 N	50–80 kg	70 ms	Strain sensor	[37]
MeMS	PEG-DA/ceA	12 kHz/pH	pH 3–8	-	pH sensor	[52]
PDMS/PVDF	Acetone/DMF	15.68 kPa^−1^	1.3–2 kPa	66 ms	Monitor human motion	[53]
ZnO	PDMS/(p(NVCL-co-DEGDVE)	364 pC N^−1^	V = ±20 V	10 ms	Artificial skins	[54]
PVDF/Ppy	Gelatin	32.39 kPa^−1^	0.1–55 kPa	200 ms	Strain sensor	[55]
PVDF-TrFE	CHACC	GF =19.3	5−25 Hz	63.2 ms	Motion Monitoring	[39]
PVDF-HFP-DBP	Polyacrylamide/Sodium-Alginate	7.7 Pa^−1^	50 kPa	-	Epidermal device	[56]

BCE: biological carbon electrodes, PEG-DA: poly(ethylene glycol) diacrylate, CHACC: cross-linked chitosan quaternary ammonium salt.

**Table 2 micromachines-14-00167-t002:** Hydrogel-based piezoelectric devices for energy harvesting applications, different piezoelectric materials and hydrogels, the critical parameters, and related performance.

Piezoelectric Materials	Hydrogel	Voltage	Current	Cycle Number	Performance	Ref.
BaTiO_3_	bacterial cellulose	14 V	1.9 µA·cm^−2^	3000	Output power 0.64 μW·cm^−2^ at 60 MΩ 1.5 V Peak output under cyclic bending BaTiO_3_ uniformly distributed in the hydrogel	[78]
BaTiO_3_/MWCNT	bacterial cellulose	18 V	1.6 µA·cm^−2^	-	Charges a 1 μF capacitor to 2.5 V in 80 s Sensitivity: 8.276 V·cm^2^/N Detection limit: 0.2 N/cm^2^	[79]
ZnO	silk	25 V	0.08 μA	500	Output power ~1 mW/cm^2^ at 70 MΩ Charge the capacitor for 480 s to reach 43.6 μJ Max. voltage of 22 μF capacitor is 1.1 V	[80]
PVDF	polyacrylonitrile	30 mV	2.8 μA	-	rapid response time ~31 μs maximum piezoelectric coefficient d_33_~30 pC/N	[40]
PVDF	PAAm-LiCl	2.7 V	-	-	Charge 1 μF capacitor to 2.8 V drive a calculator with a 44.7 μF for 30 s Maintains an output voltage of 2.7 V	[81]
PVDF	PAAm-LiCl	2.45 V	-	3400	Charging voltage for 0.47 μF capacitor is 0.86 V Output power of 4 μW at 2 MΩ	[82]

PAAm: polyacrylamide.

**Table 3 micromachines-14-00167-t003:** Hydrogel-based piezoelectric devices for wound healing, different piezoelectric materials and hydrogels, the methods of stimulation, and related performance.

Piezoelectric Materials	Hydrogel	Stimulation	Performance	In Vivo Study	Area	Ref.
PVDF	PDA-PAAm	ES	10 days healing Adhesiveness: 5 kPa Output voltage/current: 0.85 V/40 nA	Mouse dorsal area	Skin	[86]
PVDF	SA	ES	3D printing scaffold Dual piezoelectric response Average output current: ±1.29 μA	Mouse Skin	Skin	[89]
PVDF	carbonized PDA	ES	Hydrogel self-healing time: 8 ± 2 min Good adhesiveness: 70 ± 10 kPa Piezo-driven triboelectric material	Mouse dorsal area	Diabetic ulcers	[90]
PVDF	PAAN	ES	Large stretchability: around 380% Voltage output: 12 mV	-	-	[91]
PVDF	gellan-PVA	ES	Chitosan–gelatin-enhanced hydrophilicity of PVDF	-	-	[47]
PHBV	PDX	ES	Down-regulation of TGF-β1 and α-SMA Decreased infectious response	Mouse dorsal area	Skin	[92]
CM	PDA	ES/Photothermal	Promote the expression of Hsp90 and HIF-1α to accelerate wound healing Promoting cell proliferation and collagen deposition	Mice dorsal area	Diabetic ulcers	[93]
ZnO@graphdiyne	PDMS	ES/Nanoenzyme Catalysis	Piezoelectric potential: 28.4 mV Hybrid nanozyme kills 99.99% MRD Pathogens of MRSA and *P. aeruginosa*	Mice dorsal area	Skin	[94]
ZnO	PDMS	ES	10 days healing Output voltage: 900 mV Average pulse duration: 200 ± 78 ms	Mice dorsal area	Skin	[95]
PVDF	PVA	ES	Average output voltage: 0.32 V Heals in 12 weeks and can be partially degraded	Rabbits Osteochondral	Cartilage	[46]
PVDF	rGEL	ES/Drug	Voltage output: 35 mV Nerve repair completed in 23 days Promoting MSCs regeneration using CXCL12 and G-Exos	Rats dorsal area	Nerve	[96]

PDA: polydopamine, PAAm: polyacrylamide, SA: sodium alginate, PHBV: poly(3-hydroxybutyrate-co-3-hydroxyvalerate), PDX: polydioxanone. PVA: poly(vinyl alcohol), rGel: methacrylate gelatin gels.

**Table 4 micromachines-14-00167-t004:** Hydrogel-based piezoelectric devices for ultrasound stimulation, including different piezoelectric materials and hydrogels, the methods of stimulation, and related performance.

Piezoelectric Material	Hydrogel	Application	Stimulus	Performance	Ref.
BaTiO_3_	Chitosan	Tumor treatment	Ultrasound	BTO generates reactive oxygen under ultrasound Tumors are eradicated by piezocatalytic therapy The lifespans of all treated mice are over 40 days	[98]
BaTiO_3_	Aliginate/Pluronic	Promotes myogenic cell differentiation	Ultrasound	3D printing of skeletal muscle cell structures with BTO Promotes MYOD1, MYOG, and MYH2 expression and cell aggregation	[105]
PVDF	PMAA	Bladder PH test	Ultrasound	Hydrogel patch shrinks at PH < 7 and swells at PH > 7 Bending the hydrogel patch induces an ultrasonic response	[100]
CoFe_2_O_4_/PVDF	GMMA	Tissue stimulation	Magnetic	The magnetic field induces PVDF to promote proliferation of MC3T3-E1 Cell proliferation enhanced by around 25%.	[99]
ZnO	poly (DMA-co-MAA)	Visceral ultrasound imaging	Ultrasound	10% ZnO hydrogel as an ultrasound agent to assist in ultrasound imaging The median gray intensity reached 56 10% *w*/*w* of ZnO nanoparticles provided the highest stretchability of 260%	[101]
PVDF-TrFE	PVA	Water evaporation	Wave	0.87 V output voltage A high steam-generation rate of 2.702 kg/m^2^ h under kW/m^2^ 23% improvement in electrical activation	[102]

GGMA: methacrylated gellan gum, PMAA: poly(methacrylic acid-co-acrylamide), DMA: N,N-dimethylacrylamide-co-maleic acid.

## Data Availability

Not applicable.

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
