# Peer review of "Recent Progress on Hydrogel-Based Piezoelectric Devices for Biomedical Applications"

_micromachines, 2023, doi:10.3390/mi14010167_

Round 1

Reviewer 1 Report

Dear authors

This is a comprehensive review related to applications of hydrogel-based piezoelectric devices in the biomedical field. Although it was well-written, I think it needs to be revised according to the following comments before publication.

1. The authors used many abbriviations without explaination for the firt appearance such as PVDF, PAN, GF, ....

2. In the introduction section, the authors should provide a short definition of piezoelectric sensors and their differnece and advatages, compared with other sensors in biomedical applications

3. The data in Table 1, 2, and 4 was just summaried in few references. They donot show generality. Please extending the data

4. The current problems and future development strategies of hydrogel-based piezoelectric sensors should be pointed to provide potential approaches for the upcoming researches 

Author Response

Please find it as attached.

Reviewer 2 Report

Manuscript title: Recent progress on hydrogel based piezoelectric devices for biomedical applications  

This study has certain significance in material/hydrogel . However, revisions are necessary for the current version of the manuscript. The following questions to be addressed/considered may be helpful to improve the manuscript.

Major comments

·       Insufficient Abstract: In the abstract, the main aim and background of the manuscript are missing, the current version it only highlights the result. In addition, it would be even better to have a sentence as a future perspective.

·       Lake of scientific literature to support the statements and findings throughout the manuscript…... I have made some suggestions for that and more need it….

·       More information is needed for ALL TABLE captions and define the abbreviation and units that are used. And adjust the significant figures for the table and manuscript.

·       I have a major concern about the results and discussion section. The authors describe the results and compare the results with previous studies, however, insight mechanisms are still insufficient.

Specific comments:

These two paragraphs belong to the introduction section, please consider rephrasing or moving the paragraph to the introduction: line 102-107, 167-171

These sections are repeating information already presented and explain things in an unnecessarily complicated way. The quality of the manuscript would benefit from the whole section being condensed, Line 174-205, Line 393-453, Line 468-502, Line 540-567.

Line 78-99: A reference is needed here, for example, you can use:  https://doi.org/10.3390%2Fgels3010006

Line 107-120: A reference is needed here, for example, you can use: https://doi.org/10.3390%2Fma12203323

Line 324-348: A reference is needed here, for example, you can use:  https://doi.org/10.1002/app.41446

Please consider adding a reference for other places..….

Reviewer 3 Report

In this manuscript, the authors reviewed the recent progress of hydrogel based piezoelectric devices in the biomedical fields. The authors explained the great mechanical properties of the hydrogel and its great potential in wearable or deformable devices in the future. The major applications explored in this field have been well summarized by the authors. The potential applications covers, pressure/strain sensing, energy harvesting, would healing, ultrasound related applications. All the figures and tables are well preapred for the readers to get the clear understanding and full pictures. This manuscript is organized in high quality. I have the following comments and recommendations that hope the authors can help address:

1. If possible, it would be great if the authors could add a section to explain the details of hydrogels. For example, what is the typical hydrogel material, what is the synthesis process, why it has supreme mechanical properties. It is a technology in its infancy. It would be great for readers to get educated and understand this technology from fundamentals.

2. The tables are well-prepared. It would be great if the authors can add the performance reference of the state of the art solutions to compare. It will give the readers a sense of the advantages/improvements for hydrogel.

3. The conclusions and outlook part is more like a discussion in what is the potential challenges of the hydrogel in various applications. This is fantastic for the readers to know. If there will be some reference talked about where we are to solve the challenges, that would be great.

Round 2

Reviewer 1 Report

It can be accepted for publication

Reviewer 2 Report

The revised manuscript has improved compared to the original version. The authors tried to address my questions as much as possible. I recommend the manuscript to be published!